# The Role of Victim’s Resilience and Self-Esteem in Experiencing Internet Hate

**DOI:** 10.3390/ijerph192013149

**Published:** 2022-10-13

**Authors:** Wiktoria Jędryczka, Piotr Sorokowski, Małgorzata Dobrowolska

**Affiliations:** 1Institute of Psychology, University of Wroclaw, ul. Dawida 1, 50-527 Wroclaw, Poland; 2Institute of Education and Communication Research, Silesian University of Technology, ul. Hutnicza 9-9A, 44-100 Gliwice, Poland

**Keywords:** Internet hate, human–computer interaction, resilience, self-esteem

## Abstract

Despite the growing prevalence of research on Internet hate, little is still known about the psychological factors that differentiate those who are negatively affected by being subjected to Internet hate and those who are not affected at all or only to a small degree. In the present studies, we aimed to verify if resilience and self-esteem could be predictors of such responses. A total of 60 public figures (politicians, athletes, and artists; 46.7% women) and 1128 ordinary Internet users (25.1% women) participated in the study. Participants completed The Brief Resilience Scale, The Self-Esteem Scale, and The Internet Hate Concern Scale, which was created for this study, and determined how often they experience hate online. The results showed that the public figures experience Internet hate more often but were less concerned with it than the ordinary Internet users, who received online hate less often, but were more worried about it. In both groups, high self-esteem and high resilience were negative predictors of greater concern with received online hate. Our study is the first step to understanding what makes the difference between people who cope well and are not particularly concerned, and people who are greatly affected by received Internet hate.

## 1. Introduction

Internet hate is not a new phenomenon. In fact, online hate evolved long before the Internet became commonly used by ordinary, everyday people. Since the first extremist group leader created a hateful computer bulletin in 1985, this phenomenon has been constantly increasing [1,2,3]. Despite the ongoing development of strategies and interventions that are meant to curtail online hate (e.g., [4,5,6]), online spaces cannot be considered hate-free. Data point to quite the opposite—the number of hate-containing comments online is increasing [7], and the Internet hate appears as much on dedicated sites [8] as on reliable channels [9]. The atmosphere of anonymity, conformity, and sense of impunity in the online environment, fosters aggressive behaviors (e.g., [10,11,12]). As online spaces are widely used by an increasing number of people worldwide, the problem is now drawing researchers’ attention (see e.g., [13,14,15] for review).

We can list a number of different types of hatred shown online: hate speech (an induction of negative attitudes toward a social group, e.g., a gender, nation, or race (e.g., [16,17,18])), cyberbullying (defined as ‘an aggressive act or behaviour that is carried out using electronic means by a group or an individual repeatedly and over time against a victim who cannot easily defend him or herself’ (e.g., [19,20,21])), trolling (often perceived as a less offensive form of cyberbullying which is used in order to provoke or annoy someone (e.g., [22,23,24])), and hating, defined as the activity of posting online an explicitly offensive negative and exaggerated judgment of a person or an object [25]. Hating is the most personal form of attack from those listed above, as it targets harming particular individuals. At first, it was common to observe hating directed toward famous people and celebrities (e.g., musicians, politicians, or athletes), but now it increasingly becomes a problem for ordinary Internet users too and applies to the majority of young social media users [26]. It is time to intervene in unsafe online spaces [27], but it would also be helpful to better protect the victims of such Internet hate.

In the context of hating, so far, most of the psychological research raises the topic of haters’ personality and individual traits (e.g., [28]). Nevertheless, there are not many psychological studies on the victims of online hating or on how people cope with the experience of becoming such a victim. In general, we know that online hatred has been shown to induce negative emotions, to cause distress and loss of confidence, and to provoke feelings of anger, shock, fear, depression, vulnerability, and anxiety, and even lead to the suicides or killings of public figures [29,30]. It is not known, however, which factors differentiate people who cope well from those who cope poorly. In a general situation of exposure to stress, some individuals have a greater ability of coping, which can minimize negative outcomes. Research on stress distinguishes at least two factors that may be responsible for better stress coping: self-esteem [31] and resilience [32,33]. We assume that the situation of handling Internet hate attacks as well can be better, thanks to some psychological factors. In our two studies, we tested if these factors could be higher psychological resilience and higher self-esteem.

Being psychologically resilient means being better able to cope with stressful situations, not letting them negatively affect one’s wellbeing and self-efficiency [34], to be optimistic [35], and to be able to bounce back from adversities [36]. Resilience is the result of a dynamic psychological process that does not eradicate the stressor but allows one to cope with the stressing situation effectively [37]. Empirical research on resilience demonstrates a positive correlation with self-esteem [38], life satisfaction [39], and better mental health and well-being [40]. 

To have high self-esteem means appraising one’s own value positively [41]. As empirical research shows, to have high self-esteem is to be optimistic [42], more satisfied with life, happier [43], and to cope better with stress [44]. What is more, self-regulatory strategies and responses to failure are more adaptive and functional among people with high self-esteem compared to those with low self-esteem [45]. Therefore, we hypothesized that higher resilience and higher self-esteem would predict lesser concern for hate on the Internet. 

To test our hypothesis, we designed two studies. In both studies, we tested whether higher psychological resilience and higher self-esteem can predict lesser concern for offensive messages on the Internet. In Study 1, we invited public figures, who are potentially exposed to Internet hate more often than ordinary Internet users, to take part. In Study 2, we invited the ordinary users of forums and social media to complete a similar survey.

## 2. Studies

### 2.1. Study 1

#### 2.1.1. Materials and Methods

##### Participants

The study was set in Poland and all the participants were Polish. Invitations for taking part in our project were sent to politicians (ministers, members of parliament, and Polish European parliament members), athletes (Olympians from Tokyo 2020 and Beijing 2022), and the unions of artists registered and widely known in Poland, with a request to send an invitation to union members (plastic artists, musicians, and scenic artists). The sample consisted of *n* = 60 (46.7% women, 53.3% men), including *n* = 35 politicians (66% men, 34% women), *n* = 19 athletes (47% men, 53% women), and 6 artists (100% women). The response rate was approximately 5–10% among the politicians and athletes, and approximately 1% among the artists.

##### Methods

Ethics committee approval was obtained at the University of Wroclaw, decision number: 2021/EDBNI. Participants were presented with a survey in which we measured their resilience, self-esteem, Internet hate exposure, and Internet hate concern. 

The Brief Resilience Scale (BRS; [46]) consists of 6 statements measuring one’s resilience, e.g., “It does not take me long to recover from a stressful event” rated by participants on a 5-point scale, from “strongly disagree” to “strongly agree”, with 3 items reverse-coded. Due to the lack of any short-form Polish adaptation of the resilience scale, we decided to back-translate the BRS into Polish. Reliability, checked with Cronbach’s alpha, was satisfactory (α = 0.76). 

Self-esteem was measured using a Polish adaptation [47] of Rosenberg’s Self-Esteem Scale (SES; [48]). The SES consists of 10 statements measuring one’s self-esteem, e.g., “At times I think I am no good at all”, responded on 4-point scale from “strongly agree” to “strongly disagree”, and 5 items were reverse-coded. Reliability was checked with Cronbach’s alpha and was satisfactory (α = 0.73). 

Internet hate exposure: Participants were first provided with a definition of “hate” (“explicitly offensive negative and exaggerated judgment of a person”) and then asked to assess “how often does hate [directed towards them] appear in the media?”, on a 9-point scale from “every day” to “never”. 

Internet hate concern: the level of concern with hate was measured with 6 questions regarding the participants’ attitude toward hate, e.g., “Offensive comments on the Internet make me feel much less valuable”. The Participants responded on a 7-point scale ranging from “strongly disagree” to “strongly agree”. The reliability, checked with Cronbach’s alpha, was again satisfactory and amounted to *α* = 0.74. These items are available in Appendix A.

##### Results

Descriptive data from both studies are presented in Table 1. In Appendix B we present data only from those participants who have reported being victims of online hate.

To determine whether higher resilience and/or higher self-esteem could predict lesser concern with received hate, a regression model was prepared. Only data from those participants who had reported having ever been victims of hate were considered (*n* = 53, 43.4% women). The model included Internet hate concern as a dependent variable, and the following independent variables: resilience, self-esteem, Internet hate exposure, age, and gender. The model fitted the data well (adjusted R^2^ = 0.41; F(5.47) = 6.44; *p* < 0.001). Only higher resilience was a negative, significant predictor of higher Internet hate concern (β = −0.41; *p* < 0.001). This means that, when controlling for age, gender, and the appearance of Internet hate, only higher resilience can predict lesser concerns with such Internet hate. The model is presented in Table 2.

Additionally, one more model was prepared to determine whether only hypothesized predictors (resilience and self-esteem) may predict Internet hate concern. The regression model was significant (adjusted R^2^ = 0.34; F(2.50) = 12.64; *p* < 0.001). Resilience (β = 0.492; *p* < 0.001), as well as self-esteem (β = 0.248; *p* < 0.05), were negative predictors of higher Internet hate concern. This means that higher resilience as well as higher self-esteem could to some extent predict lower concern with received Internet hate. An extra model is available in Appendix C.

##### Discussion

In this study, we tested whether higher self-esteem and higher resilience can predict lesser concern with Internet hate in public figures. We found that lesser Internet hate concern could be predicted by higher self-esteem and higher resilience when only these variables were included in the model as independent. Resilience remained a significant predictor after including age, gender, and Internet hate exposure. 

### 2.2. Study 2

#### 2.2.1. Materials and Methods

##### Participants

This study was also set in Poland and all participants were Polish. Invitations for taking part in our project were sent to Internet users, via Internet forums and popular websites—Polish groups on Reddit and Facebook. The sample consisted of *n* = 1128 Internet users (72.7% men, 25.1% women, and 2.2% who chose “other” or preferred not to answer).

##### Methods

Participants were presented with a very similar survey to that used in Study 1, in which we again measured resilience, self-esteem, Internet hate exposure, and Internet hate concern. This time, we did not ask about the participants’ professional field, as it was unnecessary. 

The Brief Resilience Scale [46] was assessed in the same form as used in Study 1. Reliability, checked with Cronbach’s alpha, was satisfactory (α = 0.81).

Self-esteem was again measured using a Polish adaptation [47] of Rosenberg’s Self-Esteem Scale [48]. Reliability was checked with Cronbach’s alpha and was satisfactory (α = 0.9). 

Internet hate exposure: Participants of Study 2 were also first provided with a definition of “hate” (“explicitly offensive negative and exaggerated judgment of a person”) and then asked to assess “how often does hate [directed towards them] appear in the media?”, using the same scale as the one we used in Study 1. 

Internet hate concern: the level of concern with hate was measured using the same scale as used in Study 1. The reliability checked with Cronbach’s alpha was satisfactory (*α* = 0.82).

##### Results

The descriptive data are presented in Table 1. In Appendix B, we present data only from those participants who have reported being victims of online hate.

To determine whether higher resilience and/or higher self-esteem could predict lesser concern with received hate, a regression model was prepared. Only data from participants who had reported having ever been victims of hate were considered (*n* = 508; 22.1% women). The model included Internet hate concern as a dependent variable, and the following independent variables: resilience, self-esteem, Internet hate exposure, age, and gender. The model fitted the data well (adjusted R^2^ = 0.24; F(5508) = 32.082; *p* < 0.001). Significant, negative predictors of higher Internet hate concern were resilience (β = −0.32; *p* < 0.001), self-esteem (β = −0.18; *p* < 0.001), and Internet hate exposure (β = −0.11; *p* < 0.01). Gender, coded as 0 for female and 1 for men, was a significant predictor (β = −0.13; *p* < 0.01), unlike age. This means that those more likely to be concerned with Internet hate are female users who are less resilient, have lower self-esteem, and receive hate less frequently. The model is presented in Table 3. 

Additionally, one more model was prepared to determine whether only hypothesized predictors (resilience and self-esteem) may predict Internet hate concern. The regression model was significant (adjusted R^2^ = 0.21; F(2529) = 69.53; *p* < 0.001). Resilience (β = 0.388; *p* < 0.001), as well as self-esteem (β = 0.118; *p* < 0.01), were negative predictors of higher Internet hate concern. This means that both higher resilience and higher self-esteem could predict lesser concern with received Internet hate. An extra model is available in Appendix C.

##### Discussion

In this study, we tested whether higher self-esteem and higher resilience can predict lesser concern for Internet hate in Internet users recruited from social media and forums. We found that lesser Internet hate concern could be predicted by both higher self-esteem and higher resilience when only these variables were included in the model as independent. In the model that also contained age, gender, and Internet hate exposure, only age was not a significant predictor. This means that lesser Internet hate concern can be predicted by higher resilience, higher self-esteem, and more frequent exposure to Internet hate and, moreover, more often concerns female users. 

More frequent exposure to Internet hate as a predictor of lesser hate concern may be explained by the process of adaptation—the more frequent the exposure to Internet hate, the more one gets used to it, and the less concerning it is. However, this is only the exemplary explanation and should be verified with further research. 

## 3. Additional Analyses

In an additional analysis, the samples from both studies were compared. Figure 1 presents a comparison of reported hate exposure in Public Figures and Internet Users. We observed that 48% of the public figures and only 4% of the ordinary Internet users reported being hated once a week or more often. However, over a quarter (26%) of the Internet users reported being hated a few times a year or more often.

To verify if groups differed significantly in resilience, self-esteem, and Internet hate concern, three regression models, in which those factors were included as dependent variables, were prepared. The samples differed in size, gender frequency, Internet hate exposure, and mean age; therefore, these factors were included in all the models as predictors. The models are presented in Table 4, Table 5 and Table 6.

The regression models revealed differences in the cases of resilience and self-esteem. Our groups were significantly different on this matter, even when age, gender, and Interned hate exposure were controlled. Another result was obtained in a model with Internet hate concern as a dependent variable. With the control of age, gender, and Internet hate exposure, gender and Internet hate exposure were significant predictors of the Internet hate concern, while neither age nor the sample was significant. This means that just being a public person or not is not crucial for the level of concern with online hate. 

It could also be that some other factors, not mentioned above, were influencing the differences that we observed with the simple tests. We show the comparison to point out the discrepancies between scores in our samples, but it should be noted that many other factors could have been involved in creating them. 

## 4. General Discussion

In the two studies described above, we tested whether higher psychological resilience and higher self-esteem can predict lesser concern with online hate. We have created the Internet Hate Concern scale, which allowed us to measure the declarative extent to which individuals were concerned, moved, and have doubted themselves after receiving Internet hate. The name “Internet hate concern” may not be perfect for the essence of the construct, as it seems to be related to self-worth and well-being—for example, manifested in knowing with whose opinion to worry, and which to ignore (item 4), or with self-esteem and the preservation of one’s own self-image even when someone tries to challenge it (item 3). This construct is wide, and while the scale has only six statements capturing basic sensations, it will need further research and replications to find for sure what other factors it could be related to. The proposed name of “Internet hate concern” may not be ideal, but we believe that subsequent investigations will help to improve it. 

We found that public figures, more often than ordinary Internet users, were reporting being hated online. Ordinary Internet users had lower resilience and lower self-esteem, and they were more concerned about received online hate. Additionally, in both these groups, higher self-esteem and higher resilience were predictors of lesser concern with online hate. This observation points to a very important problem—ordinary Internet users may receive hate less often than public figures, but they are more affected by it. Additionally, unlike famous people, who may be hated in the mass media (e.g., in the comments of press articles), ordinary Internet users are more likely to receive hate in a more direct way. Keeping in mind that the number of Internet users is steadily growing internationally, this problem is affecting more and more individuals. The number of people and the time they spend online—surfing on the Internet, talking on chats, scrolling social media, commenting on the content, uploading photos, and much more—is rapidly increasing. Only last year, the number of Internet users has increased globally by 4% (192 million new users in only one year [49]). At the same time, Facebook data [7] show a big rise in the number of hate-speech-containing comments reported in each three months—from 1.6 million at the end of 2017 to 17.4 million at the end of 2021.

### Limitations and Further Directions

It cannot be ignored that our studies have several limitations. First of all, no causal effect can be established with certainty. We did not conduct experiments, and therefore we cannot tell which psychological factors were causes and which were effects. On one hand, it could be that people who have been exposed to Internet hate are concerned to a small degree, and have higher resilience in the first place, but also it could be that people who learnt how not to worry about Internet hate, reinforced their resilience. Similarly with self-esteem—people who have high self-esteem may be less concerned, as they know their value, and do not allow it to be ruined by some strangers’ opinions, but it could also be that people who have learnt how to overcome negative Internet comments, strengthened their self-esteem. No cause–effect can be established with great certainty without conducting an experiment, and therefore we encourage further experimental research on this matter. 

The lesser Internet hate concern observed among public figures may as well be the process of adaptation. Humans have the ability to adapt on many levels—starting with substances (adaptation can serve as the motor of substance addiction, see [50] for review), through the senses, which, when exposed to a prolonged stimulus, can decrease sensitivity to it (so-called “sensory adaptation”; e.g., [51,52]), via cultural habits and adjusting behavior to norms (e.g., [53]) and to psychological adaptation (see [54,55]). It is then possible that public figures, who are exposed to Internet hate much more often than ordinary Internet users and receive Internet hate from a bigger media (e.g., in comments of press articles), have gotten more used to it, and learnt how to ignore it (or how not to let it affect their well-being). It is only one exemplary alternative explanation for the relations among the factors we have considered. Further research certainly could address this hypothesis. 

The response rate in Study 1 may be considered relatively low, but if we keep in mind the specifics of the sample, it does not fare badly that approximately 5% of invited politicians, athletes, and artists decided to take part in our study. The results should not be generalized to all public persons as a group, but they are the very first step to a better understanding of how to handle exposure to Internet hate. Further research is definitely required to acquire confirmation and deepen the relations we have found. Similarly, the results from Study 2 should not be generalized—the group of Internet users is enormous and differentiated, as more and more people start using the Internet every day. Our studies were conducted based on some popular Internet sites, but as the sample was not representative, no general conclusion can be driven.

The unrepresentative nature of the sample can also be recognized by the mean scores of both the self-esteem and resilience—they may seem low when compared to norms. The norm for self-esteem in the Polish standard group of similar age [47] was noticeably higher (M = 29.83, SD = 4.16) than obtained in our study (M = 26.03, SD = 7.19). Similarly with the Brief Resilience Scale—this method was not standardized in Poland, so we can compare our results (M = 2.97, SD = 0.84) to the original method’s standard group (M = 3.53, SD = 0.68) [46], and, again, our results are noticeably lower. As the group was not selected to be representative, it is possible that the websites we have reached out to and gathered our sample from were in some ways specific. However, the obtained results are not unreasonably low—similar results had been observed before in other samples (e.g., [56,57,58,59]).

Altogether, the above limitations point to the need for further research and replications so that reliable conclusions can be drawn with greater certainty.

What more can be interesting for the further research is also the environment where hate-containing comments appear. It may be that public figures who receive Internet hate via comments to press articles or other social media sites are attacked by non-anonymous or partially anonymous haters (e.g., from fake accounts), which could to some extent suppress the hater’s zeal. Ordinary Internet users, besides the possibility of receiving hate on social media, are also endangered to fully anonymous, direct attacks in games or chatrooms (e.g., [60]). It may be interesting to research the sources of Internet hate and their effect on victim’s levels of concern. 

Also, it is quite possible that public figures receive more public, un-anonymous comments than ordinary Internet users—some of them are hate comments, but some for sure are positive comments. Ordinary Internet users, who game online or chat, are more prone to receive lots of hate comments, often unbalanced with positive appraisals. What can be taken into consideration in further research is the close-up percentage of received hate comments by participant of all the comments.

All things considered, we can assume that it is higher resilience and higher self-esteem that makes public people less concerned with Internet hate, but we encourage further research and replications on this matter to establish this with certainty. 

## 5. Conclusions

In the above-described studies, we show how higher resilience and higher self-esteem can predict lesser concern with Internet hate. Public figures tend to score higher in both these factors and, therefore, are less affected by Internet hate than ordinary Internet users who are hated online less often, but when they are—they are more concerned with such attacks. With the increasing number of Internet users who can potentially be endangered by the Internet hate, we need to have these results in mind—as more and more people may experience hate online, it may be crucial to know the factors which can minimize the negative outcomes for victims.

## Figures and Tables

**Figure 1 ijerph-19-13149-f001:**
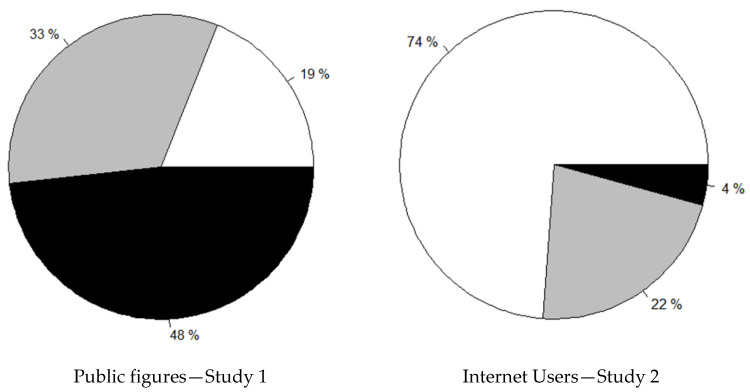
A comparison of Internet hate exposure frequency among participants in two studies. For better visibility, the data were clustered into three bundles. Chart legend: black bundle—categories from “once a week” to “every day”; grey bundle—categories from “a few times a year” to “a few times a month”; white bundle—categories from “never” to “once a year”.

**Table 1 ijerph-19-13149-t001:** Descriptive statistics of both samples.

	Age	Internet Hate Exposure (0 = Never, 8 = Every Day) *	Internet Hate Concern (1 = Very Low, 7 = Very High)	Resilience (1 = Very Low, 5 = Very High)	Self-Esteem (10 = Very Low, 40 = Very High)
Politicians (*n* = 35) Mean (SD)	51.31 (13.52)	5.38 (2.32)	2.64 (1.09)	3.46 (0.69)	34.51 (3.51)
Athletes (*n* = 19) Mean (SD)	29.17 (4.90)	3.11 (1.56)	2.54 (1.23)	3.69 (0.62)	35.63 (3.68)
Artists (*n* = 6) Mean (SD)	50.83 (12.22)	3 (3.67)	3.11 (0.81)	3.31 (0.52)	33.5 (4.89)
Total public figures (*n* = 60) Mean (SD)	44.51 (15.24)	4.43 (2.48)	2.66 (1.11)	3.52 (0.66)	34.77 (3.70)
Ordinary Internet users (*n* = 1128) Mean (SD)	23.12 (7.66)	1.36 (1.92)	3.05 (1.38)	2.97 (0.84)	26.03 (7.19)

*—The M and SD of this variable is for illustrative purposes only, as this variable is ordinal: 0 = “never”, 1 = “once in a few years”, 2 = “once a year”, 3 = “a few times a year”, 4 = “once in a month”, 5 = “a few times a month”, 6 = “once a week”, 7 = “a few times a week” and 8 = “every day”.

**Table 2 ijerph-19-13149-t002:** Regression model predicting higher Internet Hate Concern in Public Figures (*n* = 53).

Model fit measures					
Overall Model Test					
R	R^2^	F	df1	df2	*p*
0.638	0.407	6.44	5	47	<0.001
Model coefficients–Internet hate concern					
Predictor	Standardized estimate (β)	Estimate	SE	T(47)	*p*
Intercept		8.135	1.56	5.214	<0.001
Resilience	−0.406	−0.684	0.023	−3.369	<0.01
Self-esteem	−0.247	−0.073	0.037	−1.979	0.054
Internet Hate exposure	−0.177	−0.091	0.071	−1.272	0.21
Age	0.037	0.003	0.01	0.267	0.791
Gender (0 = women, 1 = men)	−0.182	−0.405	0.295	−1.373	0.176

**Table 3 ijerph-19-13149-t003:** Regression model predicting higher Internet Hate Concern in ordinary Internet Users (*n* = 514).

Model fit measures					
Overall Model Test					
R	R^2^	F	df1	df2	*p*
0.49	0.24	32.082	5	508	<0.001
Model coefficients—Internet hate concern					
Predictor	Standardized estimate (β)	Estimate	SE	T(508)	*p*
Intercept		6.046	0.306	19.738	<0.001
Resilience	−0.316	−0.514	0.076	−6.789	<0.001
Self-esteem	−0.178	−0.036	0.009	−3.753	<0.001
Internet hate exposure	−0.109	−0.083	0.03	−2.746	<0.01
Age	0.028	0.005	0.008	0.648	0.517
Gender (0 = women, 1 = men)	−0.13	−4.434	0.138	−3.131	0.002

**Table 4 ijerph-19-13149-t004:** Regression model predicting higher Resilience (*n* = 1153).

Model fit measures					
Overall Model Test					
R	R^2^	F	df1	df2	*p*
0.300543	0.090326	28.498	4	1148	<0.001
Model coefficients–Resilience					
Predictor	Standardized estimate (β)	Estimate	SE	T(1148)	*p*
Intercept		4.577	0.307	14.909	<0.001
Age	0.06	0.005	0.003	1.802	0.0718
Gender (0 = women, 1 = men)	−0.05	0.096	0.055	1.742	0.082
Group (1 = public persons, 2 = ordinary Internet users)	−0.213	−0.824	0.132	−6.26	<0.001
Internet Hate exposure	−0.274	−0.112	0.012	−9.12	<0.001

**Table 5 ijerph-19-13149-t005:** Regression model predicting higher Self-esteem (*n* = 1153).

Model fit measures					
Overall Model Test					
R	R^2^	F	df1	df2	*p*
0.554469	0.307436	127.4	4	1148	<0.001
Model coefficients–Self-esteem					
Predictor	Standardized estimate (β)	Estimate	SE	T(1148)	*p*
Intercept		52.802	2.324	22.721	<0.001
Age	0.089	0.069	0.022	3.089	<0.01
Gender (0 = women, 1 = men)	0.007	0.116	0.417	0.277	0.782
Group (1 = public persons, 2 = ordinary Internet users)	−0.387	−12.968	0.996	−13.014	<0.001
Internet Hate exposure	−0.511	−1.807	0.093	−19.519	<0.001

**Table 6 ijerph-19-13149-t006:** Regression model predicting higher Internet Hate Concern (*n* = 567).

Model fit measures					
Overall Model Test					
R	R^2^	F	df1	df2	*p*
0.279670	0.078215	11.922	4	562	<0.001
Model coefficients—Internet hate concern					
Predictor	Standardized estimate (β)	Estimate	SE	T(562)	*p*
Intercept		3.258	0.635	5.134	<0.001
Age	−0.031	−0.004	0.007	−0.614	0.54
Gender (0 = women, 1 = men)	−0.211	−0.695	0.137	−5.085	<0.001
Group (1 = public persons, 2 = ordinary Internet users)	0.077	0.377	0.255	1.476	0.140
Internet Hate exposure	−0.140	−0.102	0.031	−3.293	0.001

## Data Availability

The data are available and can be accessed: https://osf.io/s59r6/ (accessed on 7 August 2022).

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
