# Peer review of "The Role of Victim’s Resilience and Self-Esteem in Experiencing Internet Hate"

_ijerph, 2022, doi:10.3390/ijerph192013149_

Round 1
Reviewer 1 Report
The section on the regression models should have a table showing the model. Use the output from the statistical software and report regression statistics (R, standard error, etc. r2), ANOVA, regression coefficients, etc. and then discuss the table more fully in the narrative. This model is central to the findings.
Reviewer 2 Report
This is a cross-sectional study examining wether there are differences regarding Internet hate exposure, resilience, self-esteem and hate concern between public figures and ordinary Internet users. Also, the study predicts whether among those who were exposed to Internet hate, resilience and self-esteem predict internet hate concern. The study is quite simple with many constructs with questionable validity being mesured only by a single item. The topic is timely but the length of the manuscript is very short, which is evident in the depth of descriptions and discussions. There is some potential to improve the manuscript. I noticed substantial problems with the English language - a native speaker should take a closer look and correct it. Furthermore, there are comments on the content, methodological and practical level from me. More detailed comments can be found below.
Here some comments regarding the problems with the English language in the abstract. For the rest of the article I did not provide an exhaustive enumeration of language problems, as it would be too much effort to enumerate them all:
1. "those who are not so affected or only to a small degree" -> the so is redundant and should be deleted
2. "Participants completed Brief Resilience Scale, ..." -> "Participants completed THE Brief Resilience Scale, THE ...".
3. "Hate Concern Scale which created for this study" -> "Hate Concern Scale which WAS created for this study...".
Here some further comments:
INTRODUCTION
1. "It is a time to intervene in unsafe online spaces, but also helpfull would be to better protect the victims of internet hate". -> This sounds very strange and should be changed.
2. "those, who cope worse" -> this is a defining relative clause. The comma is wrong and should be deleted.
3. "to appraise own value" -> This sounds very strange and should be changed.
4. The Introduction is too short and did not acknowledge the previous knowledge relevant to the study questions. I would recommend conducting an extensive literature review, especially one that includes existing findings directly relevant to the interplay of the constructs used.
MATERIALS AND METHODS
5. The article is inconsistent in the description of the number of the studies. Whereas sometimes it is described that two studies were conducted, sometimes it is described that one study with two parts was conducted. The authors should make sure to be consistent throughout the manuscript.
6. "Peking" -> Beijing
7. What were the criteria for being considered famous as an artist? What were the fields of the artists?
8. "(... 0.81 in study 2." -> The closing bracket is missing.
9. The word "Polish" is a proper name, thus it must be written with the first letter being uppercase.
10. The measures "Hate exposure" and "Hate concern" should be named with the prefix "Internet", since they refer to Internet hate and not hate in general.
11. There are two spaces before the word "definition".
12. I am not sure if "Hate concern" is the right term for the construct. Whereas "concern" implies worry or fear, the items reads "Offensive comments on the Internet make me feel much less valuable". Thus, it seems to be more related to self-worth or self-esteem than to concern. This is a major issue that makes or breaks the study, as it calls into question the validity of the main dependent variable. It is clear that self-esteem is related to the item, since the item measures the effect on self-esteem.
RESULTS
13. There is an inconsistency in naming the construct "Internet Hate Exposure" which is sometimes also named "Hate appearance".
14. The terms "min" and "max" could be misinterpreted as empirical min and max values in the sample.
15. For the graphics do not use patterns but colors (e.g. white, medium gray, black).
16. How did you decide what answer categories to collapse/merge?
17. The charts do not include people with no exposure to Internet hate. It should be at lest mentioned in a note next to the figure caption.
18. The word "Internet" is sometimes written in lower case and sometimes with the first letter being in uppercase.
19. When comparing public figures with Internet users, there are many possible confounding variables. Therefore, it is problematic to evaluate and interpret this comparison. The authors could at least have included gender, age, etc. as a control variable.
20. Authors should explain more clearly why they used zero-truncated data for the regression analysis.
21. The text describing the regression analyses should explicitly report the sample sizes of the participants who had reported having ever been victims of internet hate.
22. A research question regarding the association between exposure and concern could be answered by using an ordinal regression analysis.
23. A moderated regression with a dichotomous variable as moderator (public figure: yes/no) could be computed instead of using two separate regression analyses.
24. The article falls short when it comes to discussing the results in reasonable depth and completely fails to point out possible limitations.
15. There are some alternative factors that might play a role that should be mentioned in the Discussion. Maybe it makes a difference, if the haters are known or are anonymous? Maybe it makes a difference what the percentage the hate comments make out in relation to non-hate comments?
16. Maybe it is an adaptation process of public figures. Maybe they started out being more concerned at the begin but then got used to it?
17. The discussion will discuss to what extent the Internet users were representative of the population of Polish Internet users. There are certainly many previous studies that show whether the values correspond to the population norms (e.g. low self-esteem or low resilience seem a bit low to me). Could there have been selection effects in the data collection here? How exactly was the data collected?
18. Another strong limitation that should be discussed is the questionable generalizability of the data, als the response rate was only 1-10%.
19. Since this is a cross-sectional study, no causal cause-effect relationship can be established. It may also be that individuals with low resilience engage in behavior that makes them more prone to online hate as a consequence. This should be kept in mind and should be made explicit.
20. There is an asterisk in the table in Appendix B at "hate appearance" but there is no explanation.
Round 2
Reviewer 2 Report
I can see that the authors took the feedback seriously and tried very hard to improve their manuscript. I appreciate this because it has noticeably improved the quality of the manuscript.
Unfortunately, there is one important point that does not allow me to recommend the publication of the manuscript in its current form: There are many problems with the English language and also with a certain imprecision concerning formal aspects. This leads to the fact that many text passages are not understandable, and therefore they can not be well assessed. I strongly and explicitly recommend that an English copy editing service be used before the next submission. This should improve the text, which currently does not meet the standards of a journal, in terms of English. Currently, many things are confusing due to the choice of wrong words and wrong word connections (collocations) and misunderstandings might occur. After using an English copy editing service, it is important to check that the original intended meaning has been retained.
A further important point: The "extra" models are important and should be presented in the main text. It is preferable to use multivariate analyses when confounding variables are not balanced across subsamples. The models without confounders are not important and should be dropped (or if you insist, should be included in the supplementary materials).
The same is true for the t-tests. They are not important - the regression analyses (equivalent to ANOVAs) are more meaningful and should be mentioned in the main text. The t-tests can be dropped or (if you insist) could be reported in the supplementary materials. If you retain the t-tests in the supplementary materials, makes sure to also report the means and SDs (or SEs) in the text.
A few more (randomly selected - there are too many errors to list them individually) points:
- "extermist group lider" - I guess you mean "extremist group leader"? Currently, it is not easy to understand what is meant here. I guess you allude to the use of dial-up computer bulletin boards by White supremacist groups?
- On some instances, you use commas instead of dots for decimal signs.
- 'strongly agree' - there is only one apostrophe in the first quotation mark
- "Resilience (β = 0.492; p < 0.001) and self-esteem (β = 0.248; p < 0.05) were positive predictors of lesser Internet hate concern." Why are you writing "positive predictor of lesser Internet hate concern?" Would it not be clearer to write "negative predictor of Internet hate concern?"
- First of all, no casual effect can be established with certainty. I guess you mean "causal" not casual".
